# Expression of Key Factors of the Hippo Signaling Pathway in Yak (*Bos grunniens*) Mammary Gland

**DOI:** 10.3390/ani12162103

**Published:** 2022-08-17

**Authors:** Peiyan Du, Bin Lu, Ling Zhao, Ying Yao, Wenjie Qian, Zhen Yu, Yan Cui, Sijiu Yu, Jiangfeng Fan

**Affiliations:** 1College of Veterinary Medicine, Gansu Agricultural University, Lanzhou 730070, China; 2Tianjin Zan Lan Technology Co., Ltd., Tianjin 300380, China; 3Technology and Research Center of Gansu Province for Embryonic Engineering of Bovine and Sheep & Goat, Lanzhou 730070, China

**Keywords:** yak, mammary gland, hippo signaling pathway, MST1, LATS1, YAP1, TEAD1

## Abstract

**Simple Summary:**

The Hippo signaling pathway plays a significant role in regulating the organ development processes of mammals. Our research aimed to investigate the expression and distribution of key members of the Hippo signaling pathway in yak mammary glands during different stages. Using immunohistochemistry, Western blot, and relative quantitative real-time polymerase chain reaction techniques, we found that the protein and mRNA expression levels of MST1, LATS1, YAP1 and TEAD1 in the yak’s mammary gland varies with the growth, lactation, and dry periods. The differential expression in the yak’s mammary gland at different stages strongly suggests that the Hippo signaling pathway plays an important role in regulating the mammary gland development processes under different physiological conditions.

**Abstract:**

Due to its rich nutritional value, yak milk is an important food source in the alpine pastoral areas. However, yaks have a low milk yield. The Hippo pathway participates in cell proliferation and organ development. We aimed to determine the regulatory mechanism of this pathway in yak mammary cells. A greater understanding of how the expression of its essential genes influence the reproductive cycle could lead to improvements in lactation performance. The expression levels of the key genes MST1, LATS1, YAP1, and TEAD1 were detected by quantitative real-time PCR, Western blotting, and immunohistochemistry during the growth, lactation, and dry periods (GP, LP and DP, respectively). The MST1 and LATS1 mRNA and protein expression level was highest during GP and lowest during LP. The YAP1 and TEAD1 mRNA and protein expression level decreased from GP to LP and DP. MST1 and LATS1 were expressed in the cytoplasm whereas YAP1 and TEAD1 were expressed in the nucleus and cytoplasm, respectively. The differential expression of MST1, LATS1, YAP1, and TEAD1 expression in the yak mammary gland during different developmental stages strongly suggests that they play an important role in the regulation of developmental functions under different physiological conditions.

## 1. Introduction

The Hippo signaling pathway contains genes that are important regulators of the size of organs; it is named after the *hpo* (Hippo) gene. This key gene of the Hippo signaling pathway was first observed in the *Drosophila melanogaster* genetic experiment in 2003; mutation of *hpo* can cause tissue overgrowth [1,2,3,4,5]. The core members of the Hippo signaling pathway have evolutionary conserved functions in the regulation of organ size [4,6,7] and have been confirmed in mice [8,9,10,11]. Currently, the generally recognized mammalian Hippo signaling pathway comprises the core factors MST1/2, LATS1/2, transcriptional coactivator YAP, and transcriptional factor TEADs [12,13,14,15,16,17,18,19]. Protein kinases encoded by these key genes form a signal transduction chain from the cytoplasm to the nucleus [20].

The Hippo signaling pathway plays a role in intracellular signal transduction and is regulated by external signals, such as growth signals. It is generally believed that the normal Hippo pathway exists in two states: on or off. When an external signal is sensed, the pathway is on, and MST1/2 is phosphorylated, causing its activation. MST1/2 in turn activates LATS1/2 via phosphorylation [21], which then phosphorylates the transcriptional coactivator YAP [22]. The phosphorylated form of YAP is retained in the cytoplasm, therefore reducing its activity in the nucleus [23]. When the external stimulation is removed, the pathway is off; YAP enters the nucleus and interacts with the transcription factor TEAD, activating gene transcription [24,25,26]. The inactivated pathway can promote cell and tissue proliferation, playing an important role in organ development (Figure 1).

The role of key members of Hippo signaling pathway in promoting proliferation and growth has been widely reported in different tissues and cell types [27,28,29,30,31]. The functions of, and mechanisms underlying, this pathway vary between environments and physiological states [29]. Recent studies have found that some variants of the Hippo signaling pathway are activated and participate in the development of mammalian reproductive organs [32,33,34,35,36,37]. Our previous study found that differentially expressed genes in the yak’s mammary tissue during lactation and dry periods were significantly enriched for the Hippo signaling pathway [38]. However, the expression of key members of the Hippo signaling pathway in mammary glands and their specific roles at different times of the yak’s reproductive cycle have not yet been examined.

Yak (*Bos grunniens*) is a livestock species found in the Tibetan plateau area, 2000–5000 m above sea level. Because of their adaptability to the area’s harsh conditions, such as high altitudes and hypoxia, yaks have become the preferred livestock in this region. The nutritional value and fat content of yaks’ milk is higher than that of cows’ milk, making it a vital material resource for the local people. The taxing environment lowers the reproductive efficiency of the yak, the majority only giving birth once every two years or twice every three years [39]. The yak’s lactation cycle is short (150 ± 30 d), the daily milk yield is low (approximately 3 kg), and the annual milk yield is approximately one seventh that of dairy cows; thus, the economic benefits are greatly diminished [40]. Therefore, studying the regulation of udder development to improve milk production, is crucial.

The mammary glands of the adult yak change regularly with repeated pregnancy. Physiological alterations, such as ductal extension, acinar proliferation and degeneration, milk production and secretion, and remodeling, occur at different stages of the reproductive cycle and are accompanied by changes in the proliferation and apoptosis of mammary epithelial cells [41]. The Hippo signaling pathway controls organ development by participating in cell proliferation and apoptosis. Therefore, studying the role and regulatory mechanism of the Hippo pathway in the yak mammary gland is a promising area of research. We hypothesized that the abundance of key factors in the Hippo signaling pathway varies in the mammary cells at different developmental stages.

## 2. Materials and Methods

### 2.1. Collection of Yak Mammary Gland Tissue Samples

All procedures involving animals were approved by the Animal Care and Use Committee of Gansu Agricultural University (GSAU-Eth-VMC-2022-024). The samples for the experiment were divided into three groups with four samples per group, and the samples were collected by simple random sampling. Twelve female yaks, with no visible pathological defects, were selected as sample individuals in the slaughterhouse of Lin Xia City, Gansu Province. The yaks were sacrificed by bloodletting from the neck, placed on their dorsal side and the size and shape of the teat were measured and recorded. A medial abdominal incision was made to open the abdominal cavity and the size and shape of the uterus and intrauterine fetus, if present, was measured. Samples of the teat parenchyma were collected, rinsed with normal saline, and divided into two processing groups. In the first group, 1-cm^3^ tissue samples were fixed in 4% paraformaldehyde solution for a minimal period of two weeks before being subjected to experimentation. In the field, the second group of samples were wrapped in tin foil and snap frozen in liquid nitrogen. In the laboratory, the samples were placed at −80 °C until required.

The developmental stage of the mammary gland can be categorized into the growth period (GP), lactation period (LP), and dry period (DP). Based on stage of the postmortem mammary gland and fetus presence in the uterus the following test groups were established: GP—medium teat size, small volume milk exudation, and fetal presence; LP—large teat size, large milk exudation volume, and no fetal presence; DP—smaller teat size, no milk exudation, and no fetal presence. The sample size of each group and crown–rump length for each fetus are shown in Table 1.

### 2.2. Extraction of Total RNA and qRT-PCR Detection

The total RNA of the 12 tissue samples was extracted using the TRIzol kit (15,596-026, Thermo Fisher Scientific, Waltham, MA, USA), according to the manufacturer’s instructions. The extracted RNA was eluted with 50 μL of RNA solution (15,596-026, Thermo Fisher Scientific, Waltham, MA, USA). The quality and quantity of the RNA in the samples was evaluated by measuring the absorbance in a spectrophotometer. All the extracted RNA samples passed quality control with OD260/280 values ranging between 1.8–2.0. cDNA for quantitative real-time PCR (qRT-PCR) was obtained by reverse transcription of the total RNA using the Reverse Transcription System (A3500, Promega, Madison, WI, USA), according to the manufacturer’s instructions.

For relative quantitative analysis of mRNA expression, MST1 (NM_001075677.2), LATS1 (NM_001192866.1), YAP1 (XM_024975708.1), and TEAD1 (XM_024975797.1) primers were designed, according to bovine sequences, and β-actin (DQ: 838049.1) primers were designed, based on yak sequences. Detailed information regarding the primers used is presented in Table 2.

We used qRT-PCR to detect the relative mRNA expression of the target and internal reference genes in each group of samples. We performed qRT-PCR with a reaction volume of a 20 μL, which included 10 μL GoTaq^®^ Master Mix (A6001, Promega, Madison, WI, USA), 0.8 μL forward and reverse primers, a total of < 1000 ng of cDNA template, and RNase-free water (to achieve a volume of 20 μL) (10,977-015, Thermo Fisher Scientific). The amplification reaction was performed using a real-time fluorescence quantitative polymerase chain reaction system (LightCycler^®^ 480, Roche, Basel, Switzerland). The threshold was determined and the CT value was calculated automatically using Light Cycler software (Roche). To eliminate technical errors, each reaction was repeated three times. The relative gene expression level was calculated according to the ΔΔCT method and expressed by 2^−ΔΔCT^ value [42].

### 2.3. Total Protein Was Extracted and Detected by WB

The tissue samples stored at −80 °C were ground to powder using a pestle and mortar with liquid nitrogen. The total protein was extracted using radioimmunoprecipitation assay buffer (R0010, Solarbio, Beijing, China). The protein concentration was quantified using a BCA Protein Assay Kit (23,225, Thermo Fisher Scientific), according to manufacturer’s instructions.

For the Western blot (WB), equal amounts of proteins were separated by 6% and 12% SDS-polyacrylamide gel electrophoresis (XP00100BOX, Thermo Fisher Scientific) and transferred to polyvinylidene difluoride (PVDF) membranes (10,600,038, Amersham, NJ, USA). The PVDF membranes were then blocked using 5% skimmed milk (9999, Cell Signaling Technology, Danvers, MA, USA) for 2 h at 25 °C before incubation with the primary antibodies against MST1 (1:1000, bs-3504R), LATS1 (1:1500, bs-2904R) and YAP1 (1:1500, bs-3605R, Bisso, Beijing, China), TEAD1 (1:1500, Ab-DF3141, Affinity Biosciences, Shanghai, China), and β-actin (1:1000, 4970, CST, Danvers, MA, USA) at 4 °C for 8 h. The PVDF membranes were then washed in PBS buffer (70,011-044, Thermo Fisher Scientific) six times (10 min, each), incubated with a horseradish peroxidase (HRP)-conjugated goat anti-rabbit secondary antibody diluent (HS101, TransGen Biotech, Beijing, China) for 1 h, and then washed with PBS buffer three times (10 min, each). The membrane was then incubated with an ECL substrate kit (AB65623, Abcam, Cambridge, UK). In the negative control group, the primary antibody was replaced with 1:400 diluted normal rabbit IgG (2729, Cell Signaling Technology). The relative intensities of MST1, LATS1, YAP1, and TEAD1 protein bands were normalized to that of β-actin.

### 2.4. Preparation of Paraffin Sections and Immunohistochemical Localization

The 1 cm^3^ samples, previously fixed with 4% paraformaldehyde, were embedded in paraffin wax. Tissue sections (thickness, 4 μm) were prepared and immunohistochemistry performed to localize the expression of the target protein.

Paraffin sections were dewaxed and rehydrated before immunohistochemical staining. First, the mammary gland tissue sections were soaked in graded concentrations of ethanol and xylene solutions before incubating in 3% hydrogen peroxide for 5 min to eliminate endogenous peroxidase activity. To enhance antigen activity, sections were heated for 15 min in 10 mM citrate buffer, while 800 W microwave irradiation was applied. After washing with PBS buffer, the sections were incubated with normal goat blocking serum (Reagent A of the Kits) for 15 min to reduce non-specific binding of the primary antibody, according to the Histostain^TM^–Plus Kits (sp-0023, Bisso, Beijing, China) instructions. Subsequently, sections were incubated with 1:500 diluted primary antibodies for 8 h at 4 °C, using the same antibodies that were used for Western blotting. After washing with PBS, the sections were incubated with biotinylated anti-rabbit secondary antibodies (Reagent B of the Kits), and then with streptavidin-conjugated horseradish peroxidase solution (Reagent C of the Kits) for 10 min at 37 °C. 3, 3-diaminobenzidine (TL-060-HD, Thermo Fisher Scientific) was used for detection before the sections were counterstained with hematoxylin, dehydrated, cleared, and mounted with resin. We also conducted a negative control experiment in which the primary antibody was replaced with a normal non-immune IgG antibody (2729, Cell Signaling Technologies). Images were captured using an Olympus-DP73 optical microscope (Olympus, Tokyo, Japan).

### 2.5. Data Analysis

The qRT-PCR data were analyzed using the 2^−ΔΔCT^ method. The relative expression of the target gene showed multiple changes relative to GP. Multiples were calculated using 2^−ΔΔCT^, where [42]:ΔΔCT = (CT _target_ − CT _internal reference_) × (CT _target_ − CT _internal reference_) _GP_(1)

In WB, the relative abundance of the target protein is presented as fold change relative to the internal reference protein.
Relative abundance = Expression of period X _target_/Expression of period X _internal reference_(2)

All data are presented as mean ± SE. The significance of the differences in the mean values between different reproductive stages of the same gene was evaluated using one-way analysis of variance, followed by Least Significant Difference. Statistical analysis was performed using SPSS 23.0 (SPSS Inc., Chicago, IL, USA). A value of 0.01 < *p* < 0.05 was considered statistically significant, while *p* < 0.01 was considered to be extremely significant.

## 3. Results

### 3.1. mRNA Expression of MST1, LATS1, YAP1 and TEAD1

Varying levels of mRNA expression were observed between the developmental stages, GP, LP, and DP. MST1 and LATS1 mRNA expression showed significant differences between the GP, LP, and DP; expression was highest in GP, followed by DP, and lowest in LP. The mRNA expression of YAP1 and TEAD1 decreased significantly between GP to LP and showed a further significant decrease in DP (*p* < 0.01) (Figure 2).

### 3.2. Expression of MST1, LATS1, YAP1 and TEAD1 Proteins

WB results showed that the β-actin protein was stably expressed in mammary tissues at all developmental stages. Statistical analysis showed that the expression characteristics of MST1, LATS1, YAP1, and TEAD1 proteins were similar to those of the mRNA expression levels. The expression of MST1 and LATS1 proteins was highest in GP, followed by DP, and lowest in LP (*p* < 0.01). The expression of YAP1 and TEAD1 proteins decreased gradually from GP to LP, and then to DP (*p* < 0.01) (Figure 3).

### 3.3. Immunolocalization of MST1, LATS1, YAP1 and TEAD1 in the Female Yak’s Mammary Gland

During the GP, the four proteins were concentrated in the alveolus clusters and ductal system and showed a strong positive signal. During the LP, the mammary gland was fully developed, the alveolus structure was clear, and the cell function was complete; MST1 and LATS1 proteins were localized in the mammary epithelial cytoplasm, while YAP1 and TEAD1 proteins were observed in the nucleus and cytoplasm of the mammary gland epithelium. During DP, the mammary gland degenerated, most of the epithelial cells were cleared after apoptosis, the parenchyma atrophied, and the stroma filled the whole mammary gland. However, part of the ductal system remained, and a positive signal appeared in the ductal system, but the intensity was reduced (Figure 4).

## 4. Discussion

In this study, qRT-PCR, WB, and immunohistochemistry identified differences in the expression intensity and distribution of MST1, LATS1, YAP1, TEAD1 mRNAs and proteins in the mammary tissues at different developmental stages. In general, the highest expression of these proteins and mRNAs was observed in GP; with expression levels decreasing during LP and DP. The histomorphology and physiological function of the mammary gland was altered at different developmental stages.

To better study the molecular mechanism of mammary gland development, the process was divided into three stages, according to the physiological characteristics: growth period (GP), lactation period (LP), and dry period (DP). Traditionally, hormones mainly regulate mammary tissue development and mammary parenchyma cells undergo physiological alterations, such as proliferation and apoptosis, induced by reproductive hormones. Although these reproductive hormones play an important role as external signals, the mechanism by which this is achieved is not fully understood.

Many previous studies have chosen a general Hippo signaling pathway model, in which the nuclear activity of YAP is regulated through the phosphorylation of MST1/2 and LATS1/2. The nuclear action of YAP transmits information from the cytoplasm to the nucleus, thereby regulating cell growth, development, and organ volume [21,22,23,24,25,26]. As discussed in this study, external signals may regulate mammary tissue development through the Hippo signaling pathway; therefore, it is crucial to explore the role of its key members in mammary tissue development at different stages. Several studies have generated mice with some of the pathway’s key genes knocked out but failed to connect these proteins to a physiologically related signaling cascade. Previous studies have established YAP as the functional output monitoring point of the Hippo signaling pathway [43,44,45,46]. Here, we associated MST1 and LATS1 with YAP1 phosphorylation and TEAD1 and explored the function of YAP in mammary gland development.

YAP is the effector of the Hippo signaling pathway and it exerts a proliferation-related phenotype by binding to transcription factors TEADs [47,48,49,50]. YAP has been shown to promote mammary alveolar cell proliferation in pregnant mice and its deletion leads to a significantly reduced acinar structure [2], and the proliferation of breast epithelial cell lines was enhanced by YAP overexpression. Furthermore, Wang et al. used immunohistochemical techniques and identified YAP expression in the cytoplasm and nucleus of luminal epithelial and myoepithelial cells from the human mammary gland [31]; similar results were observed in mouse mammary glands [35]. Our localization results from the mammary glands of yaks are in agreement with the findings from these previous studies. These in vivo and in vitro experiments demonstrate that YAP activation promotes cell proliferation and organ development. The growth rate of the mammary gland accelerates after pregnancy and is characterized by the rapid proliferation of mammary epithelial cells. In this study, the difference in expression of YAP1 and TEAD1 at the transcriptional and protein levels during GP was highly significant, suggesting that it may promote the proliferation of yak mammary epithelial cells. The mammary gland was fully developed on entering the LP, mammary epithelial cells began to synthesize and secrete milk and the cell number was no longer increasing. At the end of LP, most epithelial cells underwent apoptosis and mammary glands were remodeled. Compared with GP, the YAP1 and TEAD1 mRNA and protein were significantly downregulated during LP and their expression was lowest during DP; suggesting that YAP1 and TEAD1 may play a role in promoting proliferation and inhibiting apoptosis during mammary gland development.

There is additional evidence supporting the hypothesis that the Hippo signaling pathway mediates cell proliferation by regulating YAP activity. Studies have shown that MST1/2-induced YAP activation is an important mechanism for hepatocyte proliferation and liver volume enlargement [8,9]. Deletion of MST1/2 leads to liver enlargement accompanied by an increase in the level of YAP protein, a decrease in the level of phosphorylation, and an increase in the nuclear localization of YAP in hepatocytes. Additionally, the same phenomenon was observed in the kidney where the activation of YAP induced by MST1/2 deletion causes renal proliferation [51]. Experimental studies showed that in the intestinal epithelial cells of MST1/2-deficient mice, the degree of phosphorylation of YAP1 decreased, its abundance increased, and nuclear localization was observed. In pulmonary epithelial cells, there is a conservative MST–YAP signaling axis negatively regulating proliferation. In general, the absence of MST1/2 can promote cell proliferation in vivo and in vitro and this effect is mediated by the negative regulation of YAP activity. During breast development, MST1 may promote the proliferation of breast epithelial cells by negatively regulating YAP1.

LATS1 is widely expressed in tissues [52,53,54,55], with its highest expression in adult ovaries [6]. In vitro studies have found that excessive proliferation of ovarian somatic cells, caused by LATS1 deletion, may cause ovarian cyst formation [32]. Another in vivo study showed that LATS1 knockout led to ovarian stromal cell tumors and soft tissue sarcomas in 4-month-old mice; and mice with LATS1 gene deletion showed pituitary hyperplasia [56]. In this study, we found that LATS1 was expressed in the cytoplasm of the yak mammary gland epithelium and was significantly upregulated in the GP. It is important to note that the deletion of LATS1 leads to mammary gland hypoplasia in mice, indicating that it is necessary for the development of mammary glands and that its deletion does not cause excessive proliferation of the mammary gland. It has been suggested that excessive cell proliferation and tissue proliferation caused by LATS1 deletion are tissue-specific. Previous studies reported that intercellular contact can regulate the nuclear localization of YAP by mediating phosphorylation of LATS1/2 [10,45]. Therefore, it is possible that LATS1 is involved in mammary gland development by regulating the activity of YAP1.

In mammals, YAP is the output point of the Hippo signaling pathway. The Hippo signaling pathway regulates YAP activity mainly through the YAPS127 site. In HEK293 [10] and HPNE [57] cells, it was observed that the expression of MST1/2 and LATS1/2 and the co-expression of MST1/2 and LATS1/2 could promote YAP phosphorylation through the S127 site. The degree of YAP phosphorylation in the MST1/2 and LATS1/2 co-expression groups was more significant than that in the MST1/2 and LATS1/2 co-expression groups, LATS1/2 kinase death effectively inhibited MST1/2-induced YAP phosphorylation [52]. This suggests that LATS1/2 promotes the phosphorylation of YAP induced by MST1/2. It is suggested that breast development may be regulated by MST1-LATS1-YAP1-TEAD1 signal axis.

We found that MST1, LATS1, YAP1, and TEAD1 were all significantly upregulated in the GP mammary gland, and the cells proliferated most rapidly in this stage. However, previous studies have shown that the activation of MST1 and LATS1 induces the denucleation of YAP, which in turn inhibits cell growth. In the mammary glands of LATS1 knockout transgenic mice, it was found that the mammary epithelial tissue was absent and showed an undeveloped state, demonstrating that it is necessary for breast cancer development. However, it is still difficult to explain the significant upregulation of LATS1 during growth because the activation of LATS1 induces the inactivation of YAP phosphorylation and the significant nuclear localization of YAP promotes the proliferation of mammary epithelial cells during GP. This suggests that MST1 and LATS1 may also be involved in other pathways, and that there are additional pathways that regulate them. It is not clear how the activation of MST1 and LATS1 during mammary tissue growth leads to epithelial cell proliferation and further work is needed to answer this question. The role of the MST1–LATS1–YAP1–TEAD1 signaling axis in mammary tissue development is much more complex than suggested by this model.

In this experiment, the expression of YAP1 protein and mRNA in mammary glands showed a developmental cycle, similar to TEAD1 expression. MST1 and LATS1 showed the opposite trend with YAP1 during LP and DP, suggesting that MST1 and LATS1 may play a role in mammary glands through negative regulation of YAP. GP has the same trend, which may be the result of the influence of the collateral pathway on it, but there is also the possibility that the total protein levels of MST1 and LATS1 increase, while the level of phosphorylation decreases, due to MST1 activating LATS1, which in turn regulates the activity of YAP1 by phosphorylation. In this experiment, the expression of YAP1 protein and mRNA in mammary glands showed a developmental cycle similar to TEAD1 expression. MST1 and LATS1 expression showed the opposite trend to YAP1 during LP and DP, suggesting that they may play a role in mammary glands through negative regulation of YAP. In GP, MST1, LATS1, YAP1, and TEAD1 all showed the same trend which may be the result of the influence of a collateral pathway. There is also the possibility that the total protein levels of MST1 and LATS1 increase, while the level of phosphorylation decreases due to MST1 activating LATS1, which in turn regulates the activity of YAP1 by phosphorylation.

## 5. Conclusions

The present study is the first to establish the expression profiles of key proteins and mRNA in the Hippo signaling pathway in the mammary glands of adult yaks. We determined the relationship between the mRNA and protein expression levels of MST1, LATS1, YAP1, and TEAD1 in the mammary gland and the changes in mammary gland growth, lactation, and dryness periods. The variation in MST1, LATS1, YAP1, and TEAD1 expression in the yak mammary gland during different developmental stages strongly suggests that they play an important role in the regulation of developmental functions under different physiological conditions.

## Figures and Tables

**Figure 1 animals-12-02103-f001:**
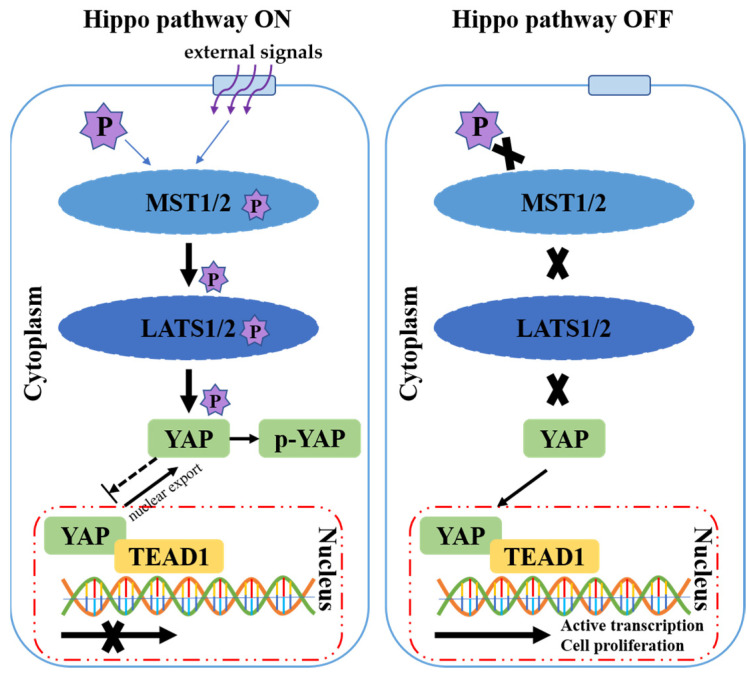
Schematic representation of the Hippo signaling pathway. Left panel: when external signals are received, the Hippo signaling pathway is switched on. Phosphorylation of MST1/2 causes activation; the activated MST1/2 phosphorylates LATS1/2 and the activated LATS1/2 phosphorylates YAP. *p*-YAP is retained in the cytoplasm and the YAP in the nucleus is inactive. Right panel: when the external stimulation is removed, the Hippo pathway is switched off. YAP is not phosphorylated and can shuttle to the nucleus and interact with TEAD1 to induce the transcription of target genes and promote cell proliferation.

**Figure 2 animals-12-02103-f002:**
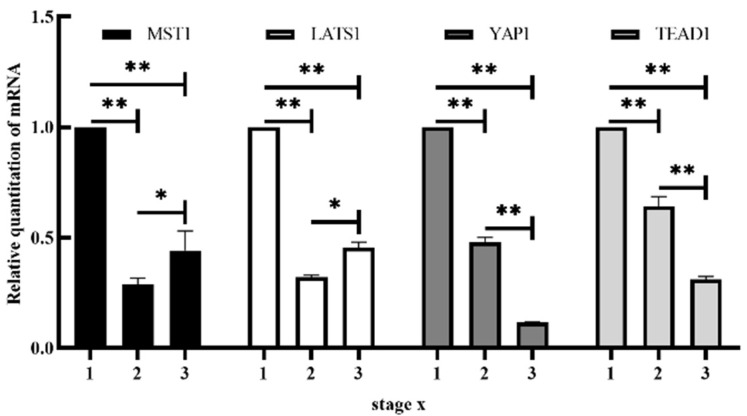
Expression of MST1, LATS1, YAP1 and TEAD1 mRNA in mammary gland of Yak at different developmental stages: 1, GP; 2, LP; 3, DP. mRNA expression levels of MST1, LATS1, YAP1, and TEAD1 showed significant variation at the different stages after normalization to the internal reference gene (β-actin). *, the expression of the same factor was significantly different in different periods (0.01 < *p* < 0.05); **, the difference was extremely significant (*p* < 0.01).

**Figure 3 animals-12-02103-f003:**
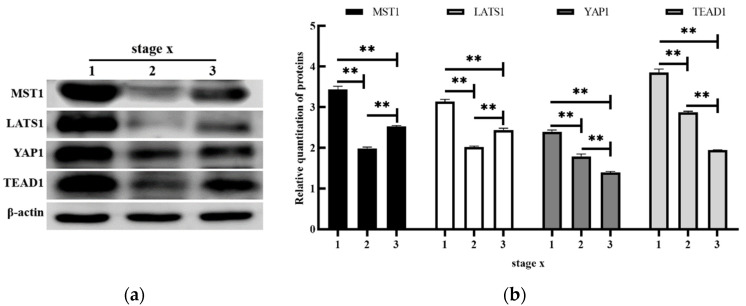
Immunoblotting bands (**a**) and statistical analysis (**b**) of MST1, LATS1, YAP1, TEAD1, and β-actin for: 1, GP; 2, LP; 3, DP. The relative abundance of target proteins at each stage was expressed as the ratio of the measured intensity of MST1, LATS1, YAP1, TEAD1 protein, and β-Actin protein (target protein/β-actin). **, the expression of the same protein was significantly different at different periods (*p* < 0.01).

**Figure 4 animals-12-02103-f004:**
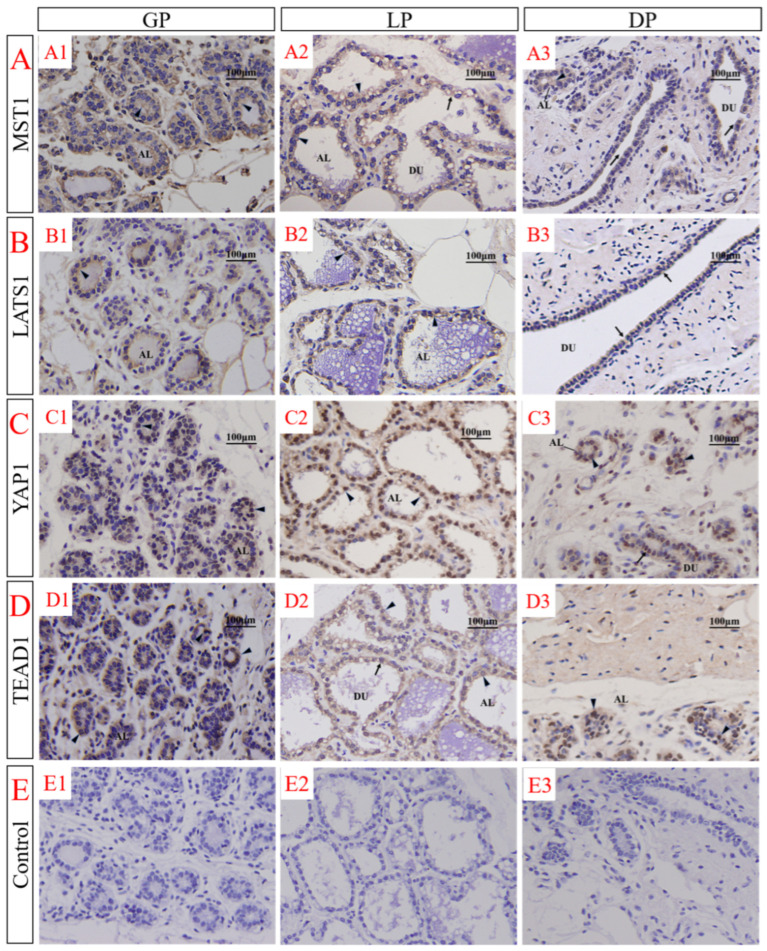
Immunolocalization of MST1, LATS1, YAP1 and TEAD1 in the female yak’s mammary gland tissues at different developmental stages. Immunolocalization of MST1 (**A**), LATS1 (**B**), YAP1 (**C**), and TEAD1 (**D**) proteins in mammary gland tissues of female yaks during GP (**A1**–**D1**), LP (**A2**–**D2**), and DP (**A****3**–**D****3**). MST1 (**A1**), LATS1 (**B1**), YAP1 (**C1**), and TEAD1 (**D1**) appeared in the alveolar clusters (AL) and ductal system (DU) during GP. During lactation, alveolar and ductal epithelial cells were positive. MST1 (**A2**) and LATS1 (**B2**) were only positive in the cytoplasm, whereas YAP1 (**C2**) and TEAD1 (**D2**) were expressed in the nucleus and cytoplasm. During the dry period, the target proteins (**A****3**,**B****3**,**C****3**,**D****3**) was mainly expressed in the ductal system and scattered in the mammary stroma. Negative control (**E**): mammary gland tissue sections were incubated with an equivalent non-immune IgG agent instead of a rabbit polyclonal antibody to MST1, LATS1, YAP1, and TEAD1 (**E1**–**E3**).

**Table 1 animals-12-02103-t001:** Number of yaks and crown–rump lengths in each group.

Group	Number of Yaks (n)	Crown–Rump Length (cm)	Characteristics
GP	4	20, 26, 28, 30	low milk exudation
LP	4	none	large milk exudation
DP	4	none	no milk exudation

**Table 2 animals-12-02103-t002:** Primers used for qRT-PCR.

Primer	Forward or Reverse	Sequence (5′–3′)	GenBank No.	Size
MST1 LATS1 YAP1 TEAD1 β-actin	Forward Reverse Forward Reverse Forward Reverse Forward Reverse Forward Reverse	GCTGCTTCTGATATGGTTCTC AGTCGTGTATGAGGTGAGTG TCCTCCACCACCTCTTAACAC TCTGACCGCTACCATTCTGAA GCTCTCCGTAGCCAGTTACCAA ATGCTGAGTCCGCTGTCTGTG GCCAACCATTCTTACAGTGA CTTGCGTATTCCGTCTCTAC CCGTGACATCAAGGAGAAG AGGAAGGAAGGCTGGAAG	NM_001075677.2 NM_001192866.1 XM_024975708.1 XM_024975797.1 DQ: 838049.1	245 bp 161 bp 173 bp 296 bp 174 bp

## Data Availability

All data presented in this study are available on request from the corresponding authors.

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
