# Peer review of "Expression of Key Factors of the Hippo Signaling Pathway in Yak (Bos grunniens) Mammary Gland"

_animals, 2022, doi:10.3390/ani12162103_

Round 1

Reviewer 1 Report

Line 13: specify in which stages

Line 55: add a figure/scheme of the pathway

Line 57: describe the roles

Line 58: add reference

Line 88: explain the statistical analysis according to which was chosen a sample of 12 animals

Line 108: add catalogue number

Table 2: add for each gene the amplicone size

Line 124: add catalogue number

Line 138: add catalogue number

Line 139: add catalogue number

Line 144: add catalogue number for all antibodies

Line 164: add catalogue information of NGS

Line 167: add catalogue number

Line 168: add catalogue number

Line 169: add catalogue number

Line 170: delete and counterstained with hematoxylin

Line 173: why authors did not perform a positive controls?

Line 173: add catalogue information

Line 189: delete Moreover, there were significant differences in expression during certain time periods.

REFERENCES: review references according guideline (like ref 1,2,11,12,13 etc)

Author Response

Dear Reviewer,

The manuscript has been revised, Please see the attachment.

Kind regards.

Reviewer 2 Report

Reviewer´s Comments – to Authors

The Article by Peiyan Du et al entitled “Expression of key factors of the Hippo signaling Pathway in Yak (Bos grunniens) mammary gland focused on the expression of components of the Hippo pathway in the mammary glands of the Yac at different stages of the lactation cycle. The experimental approaches addressed such components at both the mRNA (qRT-PCR) and protein (western blot, immunohistochemistry/IHC) levels. The Authors conclude on the relevance of the Hippo pathway in the lactation phenomenon in the Yac. This study complements previous data obtained in humans and in the preclinical (mouse) model and extends previous data from the Author´s lab. Within the scope of the Journal it will be important to glean relevant and solid data on different animal species. Thus, this research fits within the scope of “Animals” and can be welcome.

Main Comments:

1. The writing style in the English language requires extensive improvement, although overall the article reads well.

 2. This study is descriptive; gain- and loss-of-function experiments have not been performed, and thus cause-effect relationships could not be established. Therefore, conclusions in the Abstract (lines 30-33; transcribed further below) shall be softened – for example, the Authors should state that “the data strongly suggests that…..” instead of “implies that…”

“The variation in MST1, LATS1, YAP1, and TEAD1 expression in the yak mammary gland, during different developmental stages, implies that they play an important role in the regulation of developmental functions under different physiological conditions.”

3. The data obtained using immunohistochemistry/IHC for detection of Hippo pathway components falls short of desirable. Figure 3 is far from clear and aesthetically poor.  Positive signals and nuclear vs cytoplasmic signals cannot be seen clearly; relevant detail is lost. Close-ups of positive IHC signals shall be provided.

4. The supplier of the antibodies specific for components of the Hippo pathway (MST1, LATS1, YAP1, TEAD1) shall be provided; the quality of the reagents used is critical for the reliability of the data. In Materials and Methods the supplier seems to be “Bisso” (lines 144-145 of the MS); to the best knowledge of this Reviewer such supplier is obscure!!; if the supplier is a renown Author in the field that should be stated in the Materials-Methods section of the MS. Also, whether the same antibodies were used for western blotting shall be mentioned in Materials-Methods.

5. Discussion section is too long. Although comprehensive, it becomes tedious to read through. Finally, the best current model for understanding the data on the Hippo pathway, provided in the end of the Discussion section, should instead be mentioned at the beginning of this section. It could greatly facilitate the understanding of the findings of the Authors.

Author Response

(The authors gave the same response as above.)
